# Socio-economic and demographic determinants of fertility in six selected Pacific Island Countries: An empirical study

**Sumeet Lal** [ID]* [◉], **Rup Singh** [◉], **Keshmeer Makun** [◉], **Nilesh Chand** [◉], **Mohsin Khan** [◉]

Discipline of Economics, School of Accounting, Finance and Economics, The University of the South Pacific, Suva, Fiji

◉ These authors contributed equally to this work.

* lal_s@usp.ac.fj

**Data Availability Statement:** All dataset related to fertility analysis can be obtained from the "attach files" option uploaded under "additional information" section in Excel format.

## Abstract

In this study, we seek to perform macro analysis of fertility in a panel of 6 selected Pacific Island Countries (PICs, hereafter). The macro analysis with secondary data, mostly obtained from World Bank database, stretched over the period 1990–2019 was stacked randomly in a balanced panel set-up, within which the most preferred fixed effect model is used for multivariate analysis. Pooled OLS and Random effect estimation techniques were applied for comparing results. Categories such as women's empowerment, health, connectivity and cost of living were used to classify proxy variables as regressors for fertility determination. The results indicate variables such as contraceptive prevalence rate, female labour force participation rate and consumer price index (inflation) are negatively correlated with fertility at 1% level, while urbanisation is negatively correlated with fertility rate only at 10% significance level. Real GDP has negative relationship with fertility, however it is not statistically significant. Variables that are positively correlated with fertility but hold limited to no significance effects are female secondary enrolment, female population, mobile subscription and infant mortality rate. It is implied that those variables that are negatively associated with fertility, as well as Real GDP will be the major drivers for achieving replacement level fertility in the long run.

## 1. Introduction

Fertility is one of the three primary components of population dynamics that influence the population growth, structure, and composition. A study done by Pew Research Centre [1] suggests that global population growth is decreasing because of falling fertility rates. Global fertility rates has been declining for the past few decades and it reached historically low of 2.47 births per woman between 2015–2020 [2], and this trend is expected to persist until very few countries have high fertility rates [3].

Similarly, global diffusion process has led to continued fertility decline in the Pacific. Between 2015–2020, the region noted the lowest fertility rate of 3.46 [4]. Countries such as Fiji,

**Funding:** The authors received no specific funding for this work.

**Competing interests:** The authors have declared that no competing interests exist.

Samoa, Tonga, Vanuatu Solomon Islands and PNG have gone under massive demographic transition over the past few decades [5]. This is reflected in the fertility variables that have been continuously declining over several years, causing population growth rate to stumble (See Table 1). The Pacific region's population is just over 12.3 million and is expected to reach 13.5 million by 2025, which is not a substantial growth [6].

Pirie [7] states that there might be "hidden" explicit policies set in motion by existing state activites and by the general style of local development to influence demographic trends in the Pacific. Whilst these measures are mosty directed towards families with low income in the Pacific, evaluating the effect of specific policies on fertility in individual PICs might be difficult [8].

Given the fact that all PICs are classified as Intermediate Fertility Countries and looking at the pace at which these countries have seen a decline in their Total Fertility rate (TFR, hereafter) over the past two decades, the length of below replacement level period is most likely to be high [4]. It is also concluded that almost all PICs will reach the assumed TFR of 2.1 children per woman by 2100 (under the most plausible medium variant assumption) or by 2060 under low variant assumption [9], see Fig 1.

The country's ability to reach replacement level fertility in a shorter period depends on current TFR, existing levels of population momentum and the levels of structural change (socio-economic development) [10]. Overall, fertility varies slightly over the projection period in such a way that the net reproduction rate always remains equal to one, thus ensuring the replacement of the population over the long run [11]. Since below replacement level fertility is more apparent for transitional economies before 2100, constructing the target period when countries might achieve replacement level fertility would be useful [12]. Replacement level or low fertility would produce an age structure that creates a momentum for future population decline, particularly among the young [13]. This situation must be stopped at some point if the population is to be demographically and politically sustainable. The effect of population decline, caused by declining fertility rates would have serious demographic and socio-economic problems in the future [14]. Declining TFR can significantly affect labour force, economic productivity, and human capital formation in any country [15]. Fertility rates may be linked to the amount of money spent on women's maternal health and education, which might have a direct impact on workforce and productivty in the long run [16]. Onarheim, et al [17] state that women who are healthy contribute to societies that are more educated and more productive and ensuring women's control over their own reproduction helps accelerate economic development and prosperity. The Pacific has already started experiencing the implications of falling fertility, which is reflected in the rise in elderly population, which has started to put pressure on public health services. Report from United Nations Population Fund [18] suggests that between 2014 and 2050, the population of elderly people in the Pacific will increase at a 3.7 percent yearly, rising from about 512 thousand to 2 million. The oldest elderly (those aged 80 and above) are now growing at a higher pace than those aged 60 and above. This could make provisions of health services and long-term care for the oldest or disabled very difficult, particularly in rural areas and outer islands [19].

The potential effects of lower fertility on the human population in the Pacific may be diverse. Fertility combined with mortality and migration will affect the structure of the human population and will have negative implications for economic growth, human capital creation, elderly age dependence and health services, demographic dividend, as well as leadership and governance [20]. Generally, investments in education and vocational training are particularly important in nations with a big proportion of young people and high fertility rate at initial level. Training and health care, including sexual and reproductive health are also important to enable country to create a window of opportunity [21]. However, in the Pacific, such initiatives

**Table 1. Linkage between fertility variables and population data.**

| Country | Fertility variables | Time periods | | | | | | | |
|---|---|---|---|---|---|---|---|---|---|
| | | 1980–1985 | 1985–1990 | 1990–1995 | 1995–2000 | 2000–2005 | 2005–2010 | 2010–2015 | 2015–2020 |
| Fiji | Sex ratio | 1.06 | 1.06 | 1.06 | 1.06 | 1.06 | 1.06 | 1.06 | 1.06 |
| | Mean-age @ childbearing | 27.40 | 27.60 | 26.83 | 27.76 | 27.76 | 27.58 | 27.98 | 28.12 |
| | TFR | 3.80 | 3.47 | 3.35 | 3.19 | 2.98 | 2.75 | 2.79 | 2.79 |
| | **Population data** | *Impact of change in fertility variables on population* | | | | | | | |
| | Population density | 37.0 | 39.6 | 41.3 | 43.7 | 44.7 | 46.2 | 47.4 | 48.4 |
| | Population growth | 2.27 | 0.47 | 1.25 | 0.90 | 0.26 | 0.91 | 0.20 | 0.63 |
| | Total population(000) | 712 | 729 | 775 | 811 | 822 | 860 | 869 | 896 |
| Vanuatu | Sex ratio | 1.07 | 1.07 | 1.07 | 1.07 | 1.07 | 1.07 | 1.07 | 1.07 |
| | Mean-age @ childbearing | 29.40 | 29.36 | 29.45 | 29.54 | 29.43 | 29.32 | 29.32 | 29.28 |
| | TFR | 5.40 | 5.04 | 4.83 | 4.59 | 4.40 | 4.20 | 4.00 | 3.80 |
| | **Population data** | *Impact of change in fertility variables on population* | | | | | | | |
| | Population density | 10.1 | 11.4 | 13.1 | 14.6 | 16.3 | 18.5 | 21.1 | 24.0 |
| | Population growth | 2.35 | 2.40 | 2.75 | 1.91 | 2.47 | 2.42 | 2.76 | 2.49 |
| | Total population(000) | 130 | 147 | 168 | 185 | 209 | 236 | 271 | 307 |
| Solomon Islands | Sex ratio | 1.07 | 1.07 | 1.07 | 1.07 | 1.07 | 1.07 | 1.07 | 1.07 |
| | Mean-age @ childbearing | 29.56 | 29.39 | 29.27 | 29.15 | 29.23 | 29.43 | 28.88 | 28.88 |
| | TFR | 6.43 | 6.13 | 5.53 | 4.91 | 4.60 | 4.40 | 4.44 | 4.44 |
| | **Population data** | *Impact of change in fertility variables on population* | | | | | | | |
| | Population density | 9.0 | 10.5 | 12.1 | 14.0 | 16.0 | 18.0 | 20.4 | 23.3 |
| | Population growth | 3.22 | 2.83 | 2.83 | 2.77 | 2.60 | 2.33 | 2.67 | 2.60 |
| | Total population(000) | 271 | 312 | 359 | 413 | 470 | 528 | 603 | 687 |
| Samoa | Sex ratio | 1.08 | 1.08 | 1.08 | 1.08 | 1.08 | 1.08 | 1.08 | 1.08 |
| | Mean-age @ childbearing | 30.79 | 31.01 | 31.23 | 29.80 | 30.00 | 30.14 | 30.23 | 30.33 |
| | TFR | 5.91 | 5.35 | 4.92 | 4.62 | 4.44 | 4.47 | 4.16 | 3.90 |
| | **Population data** | *Impact of change in fertility variables on population* | | | | | | | |
| | Population density | 55.8 | 57.1 | 59.0 | 61.1 | 62.7 | 64.8 | 67.3 | 69.4 |
| | Population growth | 0.57 | 0.35 | 0.87 | 0.51 | 0.60 | 0.68 | 0.80 | 0.50 |
| | Total population(000) | 160 | 163 | 170 | 174 | 180 | 186 | 194 | 198 |
| Tonga | Sex ratio | 1.05 | 1.05 | 1.05 | 1.05 | 1.05 | 1.05 | 1.05 | 1.05 |
| | Mean-age @ childbearing | 30.80 | 30.76 | 30.90 | 31.04 | 31.07 | 31.07 | 31.08 | 31.08 |
| | TFR | 5.50 | 4.74 | 4.62 | 4.29 | 4.23 | 4.03 | 3.79 | 3.58 |
| | **Population data** | *Impact of change in fertility variables on population* | | | | | | | |
| | Population density | 130.0 | 131.4 | 132.8 | 134.8 | 138.4 | 143.2 | 141.6 | 143.5 |
| | Population growth | 0.20 | 0.25 | 0.19 | 0.41 | 0.59 | 0.60 | -0.63 | 0.95 |
| | Total population(000) | 94 | 95 | 96 | 98 | 101 | 104 | 101 | 106 |
| PNG | Sex ratio | 1.08 | 1.08 | 1.08 | 1.08 | 1.08 | 1.08 | 1.08 | 1.08 |
| | Mean-age @ childbearing | 30.12 | 29.96 | 29.90 | 29.83 | 29.76 | 29.78 | 29.80 | 29.81 |
| | TFR | 5.47 | 4.97 | 4.70 | 4.64 | 4.39 | 4.13 | 3.84 | 3.59 |
| | **Population data** | *Impact of change in fertility variables on population* | | | | | | | |
| | Population density | 8.4 | 9.7 | 10.9 | 12.3 | 13.8 | 15.4 | 17.2 | 19.0 |
| | Population growth | 2.67 | 2.46 | 2.33 | 2.40 | 2.10 | 2.37 | 2.07 | 1.97 |
| | Total population(000) | 4 081 | 4 616 | 5 187 | 5848 | 6 495 | 7 311 | 8 108 | 8947 |

United Nations Population Division. https://population.un.org/wpp/Download/Standard/Fertility/.

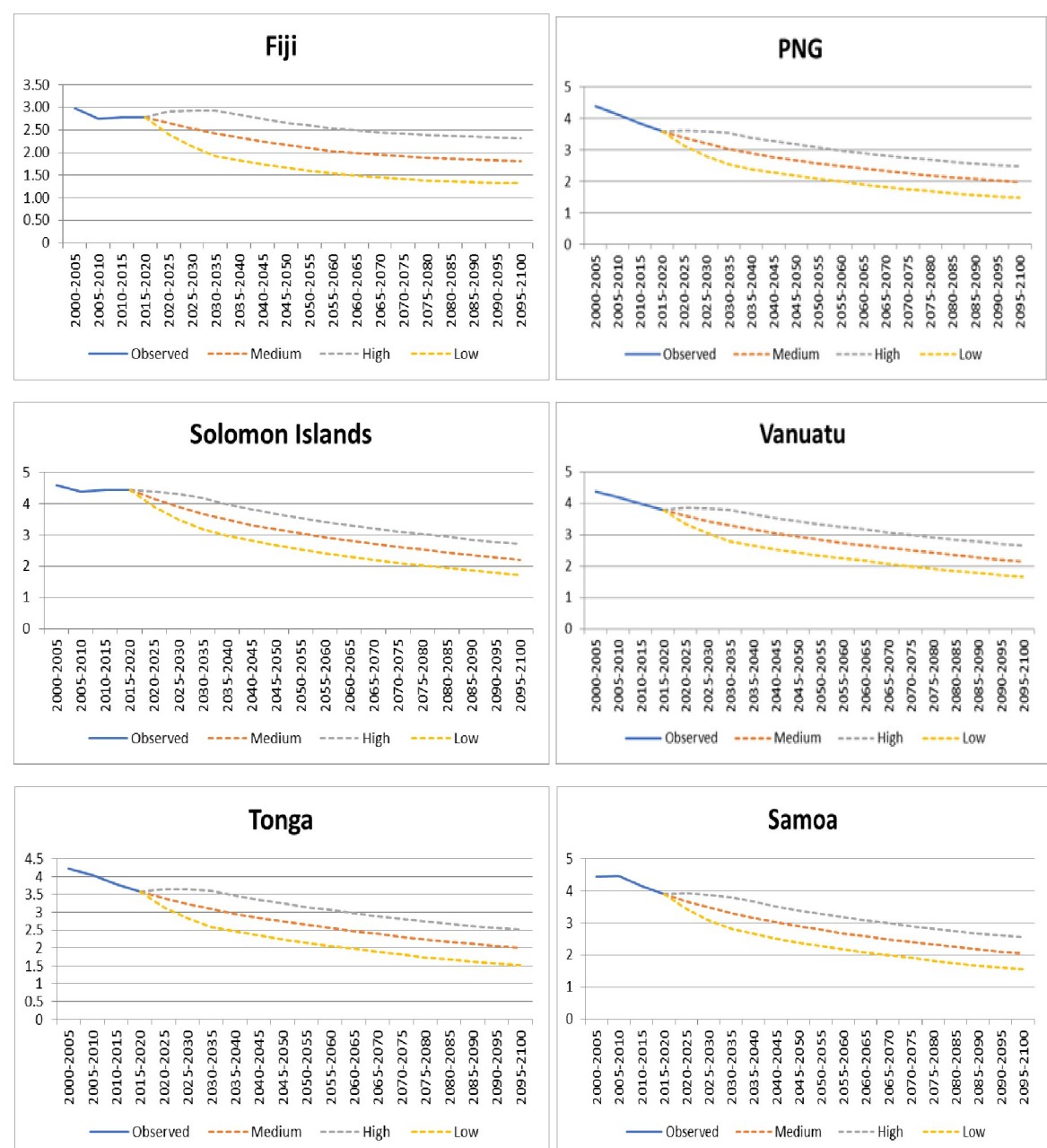

**Fig 1. Observed and projected total fertility rates for 6 PIC's until 2100.** United Nations Population Division. (2019, August 23). Total fertility by region, subregion and country. Retrieved from World Population Prospects: https://population.un.org/wpp/Download/Standard/Fertility/.

are lacking at a considerable level due to financial constraints, making it difficult for the region to reap the first demographic dividend [22]. Continued fertility decline, coupled with the low fertility rates in most PICs means that in the absence of policies seeking to manage fertility decline beyond the acceptable level, the Pacific will continue to experience decline in population growth, that will constrain development and will bypass the opportunities of demographic dividend [23]. Thus, declining population growth rate, largely triggered by declining fertility rates becomes an important topic to explore in the Pacific.

Davis & Blake [24] first proposed the first systematic classification of the proximate determinants of fertility through which economic, social and cultural factors could influence fertility. Bongaarts [25] later modified this framework and specified individual proximate determinants that could influence fertility. Davis, Blake and Bongaart's research was closely complemented by Ahlburg & Cassen [26]. Recently, new approaches, which allow investigation of these interrelationships at the individual level, have been developed Singh, et al [27]. Meanwhile, studies on determinants of fertility in Pacific has been limited and out-dated. To the best of our knowledge, only two studies done by Gani [28] and Lucas & Ware [29] has empirically explored variables influencing fertility in the Pacific. Gani [28] performed time series investigation of factors affecting fertility rates in PNG, Solomon Islands and Vanuatu using cross-country data from 1982–1993 and found that higher infant mortality rate is strongly correlated to high fertility rates. The empirical results obtained also confirm that there is a strong inverse correlation between family planning, urbanization, income, female education with total fertility rates. Lucas & Ware [29] surveyed 18 PICs from 1960s and 1970s and found that in several countries, fertility has declined significantly because of the expansion of family planning services in the 1960s and 1970s. Quite recently, a joint report from SPC & UNSW [30] theoretically examined span across Melanesia, Micronesia and Polynesia, and found diverse cultures, colonial histories, population sizes, landmass sizes, levels of social and economic development, and governance structures as variables affecting fertility level overtime.

Given the aforementioned issues, the primary goal of this study is to look at the factors that influence fertility rates on emerging Pacific islands. To achieve this, we conduct multivariate econometric analysis in a balanced panel data setting. Variable selection was based on works of Davis & Blake [24] and Bongaarts [25] model of proximate determinants of fertility, where socio-economic and demographic conditions are related to aggregate level outcomes. Variables such as women's empowerment, health, connectivity and cost of living, within which several other variables are categorised and is examined in a panel fertility model. The main advantage of using panel analysis is that it provides data accessibility and flexibility to model wide range of human behaviours [31, 32]. Our approach of using macro panel data to model the determinants of fertility is an innovation to provide empirical backing to the theoretical insights as well as to fill gaps in knowledge on the factors that drive changes in fertility rates and generate new scientific and policy-oriented knowledge on the reproductive decision-making of contemporary Pacific. Different estimation techniques helps with better understanding of fertility behaviour and our work uses modern econometric methods to model the influences of these indicators.

## 2. Data methodology and estimation

### 2.1 Data sources and limitations

Data pertaining to all the variables used in this analysis (see Table 2), except for contraceptive prevalence rate, have been obtained from World Bank's World Development Indicators Database, spanning from 1990–2019, stacked randomly in a balanced panel set-up. The Data for contraceptive prevalence has been obtained from World Health Organisation. The database provided data for each variable that were complete, consistent, and accurate and in accordance with Data Quality Assurance Framework (DQAF). DQAF is set up by World Bank as a process for evaluating data quality that combines best practices and internationally accepted statistical ideas and terminology. Hence, the variables implied in this study are legitimate to be used for econometric analysis [33]. The only data limitation is that the variables latest values were until 2019. Thus, we were not able to use the latest figures for empirical analysis. This data limitation

**Table 2. Data definition, description and sources.**

| Variable | Acronym | Description | Source |
|---|---|---|---|
| Total Fertility Rate | TFR | Total fertility rate is the total number of births per woman throughout her childbearing years (15–49) | The data is from World Bank (2021). |
| Real GDP | Real GDP | Is an inflation-adjusted measure that reflects the value of all goods and services produced by an economy in a given year, measured in constant USD$ | The data is from World Bank (2021). |
| Labour Force Participation rate | LFPRF | Labour force participation rate, female (% of female population 15+) | The data is from World Bank (2021). |
| Gross Female Secondary School Enrolment rate | GFSSER | Gross female secondary school enrolment (% of all females, gross) | The data is from World Bank (2021). |
| Female Population | FEMPOP | Percentage of total population that are female | The data is from World Bank (2021). |
| Mobile Cellular subscriptions | MOBSUB | The number of mobile cellular subscriptions divided by the country's population and multiplied by 100. | The data is from World Bank (2021). |
| Contraceptive Prevalence Rate | CPR | Contraceptive prevalence rate (median estimates, % of total population) | The data is from WHO (2019). |
| Infant Mortality rate | IMR | Is the number of deaths per 1,000 live births of children under one year of age. | The data is from World Bank (2021). |
| Urban population | UP | Refers to the number of people that are living in urban areas. | The data is from World Bank (2021). |
| Consumer Price Index | CPI | Reflects changes in the cost to the average consumer of acquiring a basket of goods and services that may be fixed or changed at specified intervals, such as yearly. | The data is from World Bank (2021). |

The World Bank. (2021, July 7). World Development Indicators. Retrieved August 30, 2021, from The World Bank: https://databank.worldbank.org/source/world-development-indicators [34].

is justifiable given the financial, human and statistical building constraints that lie within the Pacific, making it difficult to collect and compile the latest dataset. Table 2 shows the list of all variables used in the analysis, acronyms, description (measurement) and their sources.

## 2.2 Selection of the countries

Six Pacific countries namely, Fiji, Samoa, Tonga, Vanuatu, Solomon Islands and Papua New Guinea have been selected for the study. These countries have been chosen because they have gone under massive demographic transitions over the past few decades, which have eventually taken toll on population composition, in terms of changes to the share of the younger, working age and older population [35]. These countries have also experienced substantial structural

**Table 3. Basic socio-economic and demographic indicators for 6 selected PICs.**

| Country | GDP/Capita in 2020 (Constant 2010 US$) | Population in 2020 | Population Density (pple/sq.kil) (2019) | Net Migration (2017) | Life Expectancy at Birth (2019) | Human Development Index in 2018 | Population Composition (%) | | |
|---|---|---|---|---|---|---|---|---|---|
| | | | | | | | 0–14 | 15–64 | 65+ |
| Fiji | 3808.21 | 896444 | 48.95 | -31008 | 67.444 | 0.724(98/189) | 29.01 | 65.17 | 5.81 |
| Vanuatu | 2539.57 | 307150 | 24.97 | 600 | 70.474 | 0.597(141/189) | 38.41 | 57.98 | 3.60 |
| Samoa | 3738.27 | 198410 | 70.01 | -14013 | 73.321 | 0.707(111/189) | 37.19 | 57.71 | 5.08 |
| Tonga | 4903.15 | 105697 | 144.34 | -3999 | 70.907 | 0.717(105/189) | 34.76 | 59.31 | 5.91 |
| Solomon Islands | 1632.48 | 686878 | 24.41 | -7998 | 72.996 | 0.557(153/189) | 40.03 | 56.29 | 3.67 |
| PNG | 2346.80 | 8947027 | 19.91 | -3999 | 64.501 | 0.543(155/189) | 35.14 | 61.28 | 3.57 |

The World Bank. (2021, July 7). World Development Indicators. Retrieved August 30, 2021, from The World Bank: https://databank.worldbank.org/source/world-development-indicators [34].

changes in terms of human development (Real GDP, HDI and Urbanization). Therefore, it will be of interest to see how fertility responds with respect to different socio-economic and demographic determinants. Moreover, these are the major Pacific economies comprising over 80% of the region's population and have datasets on social statistics that are well developed relative to the other smaller Pacific economies [36]. Table 3 shows basic socio-economic and demographic indicators for 6 selected PICs.

## 2.3 Model specification

Given our purpose of study, we apply the following procedure to empirically estimate determinants of fertility. We draw from Davis and Blake [24], complemented by Bongaarts [25] to conduct our analysis. The interesting conceptual frameworks presented in these studies allow us to specify the following log-linear model where TFR depends on theoretically consistent determinants representing socio-economic and demographic factors. Our empirical model is as follows:

$$\ln TFR_{it} = \alpha_i + \beta_i \sum_{x-1}^{n} \ln X_{it} + \varepsilon_{it} \tag{1}$$

Where TFR is the total fertility rate in country $i$, and $X_{it}$ represent the proximate variables set that measures causes of TFR (in logs). The t stands for the time dynamics and $ln$ is the natural logs applied to relevant variables. The error term ($\varepsilon$) is assumed to follow the white noise process. The tested socio-economic and demographic variables with prior expectations (signs with TFR) are stated below.

1. Women's Empowerment:

    a. Gross secondary enrolment of females (-)

    b. Real GDP (-)

    c. Female labour force participation rate (-)

2. Health:

    a. Infant Mortality rate (+)

    b. Contraceptive prevalence rate (-)

3. Connectivity:

    a. Urbanization population (-)

    b. Mobile Cellular subscriptions (+)

    c. Female Population (+)

4. Cost of Living:

    a. Consumer Price Index (-)

Reasons why above variables has been selected for analysis lies in a literature by Davis and Blake [24] that offered a quantitative framework to decompose fertility into its proximate determinants (social and economic variables). They show that the framework is useful for performing comparative fertility analysis. Bongaarts [25] modified Davis and Blake's framework and identified a smaller set of proximate fertility determinants such as contraceptive use, induced abortion amongst others. Researchers such as Dutt & Ros [37] also used socio-

economic variables and concluded that the global diffusion process underlies a strong correlation between fertility, infant mortality, female education and contraceptive use. Researchers have also used variables such as inflation, urbanisation rate, income, the number of young women working away from home, as the key determinants of fertility [38].

## 2.4 Estimation techniques

Fixed Effect (FE), Random Effect (RE) and Pooled OLS is employed, depending on the suitability of the model, to estimate the effect of socio-economic and demographic factors on fertility rate.

Fixed-Effects (FE) is used for analysing the impact of variables that vary over time. Within an entity, FE investigates the connection between predictor and outcome variables. Individual features of each entity may or may not influence the regressors. When FE is used, it is presumed that something about the individual will influence or bias the predictor or outcome variables, and we must account for this. The assumption of a correlation between the entity's error term and predictor variables is based on this logic. The net effect of the predictors on the outcome variable is examined by removing the influence of those time-invariant features using FE. The FE model (Eq 2) also assumes that such time-invariant qualities are unique to the individual and should not be linked with other individual characteristics. Because each entity is unique, its error term and constant (which reflects individual features) should not be linked with those of other entities. If the error terms are correlated, FE is not appropriate since inferences may not be valid. The Fixed-effects regression model is presented as:

$$TFR_{it} = \beta_1 X_{1,it} + \ldots + \beta_k X_{k,it} + \alpha_i + \varepsilon_{it} \qquad (2)$$

with i = 1. . . n and t = 1,. . ., T. The $\alpha_i$ are entity-specific intercepts that apprehends heterogeneities across entities.

The Random Effects model takes that variation among entities is considered random and uncorrelated with the predictor or independent variables included in the model:

". . .the crucial distinction between fixed and random effects is whether the unobserved individual effect embodies elements that are correlated with the regressors in the model, not whether these effects are stochastic or not" [39]

Furthermore, an advantage of random effects is that time invariant variables can be captured, which in fixed effects, is absorbed by the intercept. The random effects model is:

$$TFR_{it} = \beta X_{it} + \alpha + \varepsilon_t + u_t \qquad (3)$$

with $\varepsilon_t$ = between-entity error and $u_t$ within-entity error.

Generally, RE assumes entity's error term to be uncorrelated with the regressors, which allows time-variant variables to play a role as independent variables. The question of which model (FEM vs. REM/Pooled OLS) is preferable is based on the assumption one makes about the likely correlation between the individual, cross-section, error component, $\varepsilon_t$, and the regressors. If it is assumed that $\varepsilon_t$ and the regressors are uncorrelated, REM would be appropriate, whereas if $\varepsilon_t$ and the regressors are correlated, FEM may be appropriate. To test whether fixed or random model would be appropriate, Hausman test is performed. By caution, it is necessary to test the presence of random effects by using Breusch and Pagan Lagrangian multiplier (LM) test to determine whether to accept or refuse the pooled OLS model over random effect model.

**Table 4. Descriptive statistics.**

| Statistics | Mean | Median | Max | Min | Std. Dev | Skewness | Kurtosis | Variance | Obs |
|---|---|---|---|---|---|---|---|---|---|
| TFR | 4.13 | 4.30 | 5.85 | 2.75 | 0.65 | -0.48 | 2.98 | 0.434 | 180 |
| Real GDP | 2.98e+09 | 6.96e+08 | 2.18e+10 | 2.44 | 4.75 | 2.23 | 7.33 | 2.25 | 180 |
| LFPRF | 52.40 | 45.93 | 83.11 | 31.12 | 17.47 | 0.52 | 1.90 | 305.30 | 180 |
| GFSSER | 61.83 | 66.73 | 116.32 | 8.71 | 33.41 | -0.16 | 1.53 | 1116.33 | 180 |
| FEPOP | 48.97 | 49.11 | 50.01 | 47.79 | 0.49 | -0.33 | 3.20 | 0.24 | 180 |
| MOBSUB | 27.87 | 6.42 | 120.45 | 0 | 34.15 | 0.95 | 2.78 | 1166.74 | 180 |
| CPR | 33.83 | 31.75 | 48.80 | 23.6 | 7.06 | 0.66 | 2.19 | 49.93 | 180 |
| IMR | 25.16 | 21.75 | 62.20 | 12.9 | 11.91 | 1.62 | 4.69 | 142.06 | 180 |
| UP | 247173.3 | 59659 | 1162836 | 21584 | 318088.5 | 1.35 | 3.53 | 1.01 | 180 |
| CPI | 78.96 | 77.47 | 155.99 | 16.31 | 32.10 | 0.03 | 2.09 | 1030.47 | 180 |

Estimated in Stata 13.

## 3. Results and discussion

### 3.1 Descriptive statistics and correlation matrix

Tables 4 and 5 provides a summary and correlation matrix of the main variables used in the analysis from 1990–2019 for 6 PIC's. In Table 4, demographic variables such as total fertility rate, female population, contraceptive prevalence rate, infant mortality rate and urban population on average were 4.13 births /woman, 48.97%, 33.83%, 25.16/1000 live births and 247173.3 people, respectively. Social variables such as female labour force, female secondary school enrolment and mobile subscription averaged 52.40%, 61.83% and 27.87/100 people, respectively. Real GDP and CPI index (inflation) as economic variables averaged USD$2.98b and 78.96% as price change, respectively.

Table 5 shows the possible correlation/relationship between all the major variables used in this study. The correlation analysis shows that TFR is negatively correlated with CPR, Real GDP, CPI, MOBSUB, FEMPOP, GFSSER and UP at substantive coefficient size. Variables such as LFPR and IMR are negatively correlated with TFR. The relationship of other socio-economic and demographic variables with each other is also quite interesting.

**Table 5. Correlation matrix.**

| Variables | TFR | CPR | Real GDP | CPI | LFPRF | IMR | MOBSUB | FEMPOP | GFSSER | UP |
|---|---|---|---|---|---|---|---|---|---|---|
| TFR | 1.00 | | | | | | | | | |
| CPR | -0.80 | 1.00 | | | | | | | | |
| Real GDP | -0.18 | 0.01 | 1.00 | | | | | | | |
| CPI | -0.57 | 0.43 | 0.20 | 1.00 | | | | | | |
| LFPRF | 0.49 | -0.11 | 0.03 | -0.18 | 1.00 | | | | | |
| IMR | 0.24 | -0.19 | 0.73 | -0.27 | 0.41 | 1.00 | | | | |
| MOBSUB | -0.58 | 0.46 | 0.00 | 0.81 | -0.21 | -0.31 | 1.00 | | | |
| FEMPOP | -0.39 | 0.39 | 0.12 | 0.27 | 0.13 | 0.04 | 0.31 | 1.00 | | |
| GFSSER | -0.50 | 0.16 | -0.38 | 0.24 | -0.79 | -0.73 | 0.37 | 0.11 | 1.00 | |
| UP | -0.32 | 0.16 | 0.95 | 0.16 | 0.03 | 0.76 | 0.03 | 0.17 | -0.36 | 1.00 |

Estimated in Stata 13.

**Table 6. Hausman test.**

| Panel A: Variation in Coefficients of Fixed and Random effects | | | | |
|---|---|---|---|---|
| *Variables* | *(b)* | *(B)* | *(b-B)* | *Sqrt(diag(V_b-V_B))* |
| | *Fixed* | *random* | *Difference* | *S.E.* |
| CPR | -0.0642 | -0.0356 | -0.0285 | 0.0050 |
| LFPRF | -0.0146 | 0.0095 | -0.0241 | 0.0025 |
| IMR | 0.0099 | 0.0147 | -0.0047 | 0.0075 |
| MOBSUS | 0.0007 | -0.0007 | 0.0015 | 0.0003 |
| FEMPOP | 0.0712 | -0.2035 | 0.2748 | 0.0469 |
| SSERF | 0.0042 | -0.0036 | 0.0078 | 0.0013 |
| LRealGDP | -0.1422 | -0.0314 | -0.1107 | 0.0940 |
| LUP | -0.1427 | -0.2406 | 0.0979 | 0.0732 |
| LCPI | -0.4137 | -0.1182 | -0.2955 | 0.0505 |
| Panel B: Decision on Fixed and Random effects | | | | |
| *Test Summary* | | *Chi Square Statistics* | | *P value* |
| Difference on coefficients not systematic | | 142.56 | | <0.001 |

Estimated in Stata 13.

## 3.2 Empirical estimation

**3.2.1 Fixed, random/Pooled OLS model selection.** The explanation for differences that arise based on units and time could not be investigated simply with the fixed effects model in panel data analysis. It might also be investigated using a random effects /Pooled OLS model. If unit and/or temporal effects are discovered as a consequence of panel data analysis studies, it is necessary to determine if these effects are fixed or random. The "Hausman Model Identification Test" is used to determine whether to employ a fixed effects or random effects model in panel data studies (see Table 6). The Hausman test is used to see if there is a link between the model's explanatory variables and the model's particular effects, under the assumption that the model's specific effects are random.

Fixed effects models are valid when the probability value of the Hausman test statistic is less than 0.005. The probability value of the above Hausman test statistic is less than 0.005 and it was determined that the fixed effects model was valid and most suitable for this analysis.

We then proceed with Breusch and Pagan Lagrangian Multiplier test (Table 7) to determine whether we still choose Random effect or Common/Pooled OLS effect. The null hypothesis in the LM test is that the variances across entities are zero. There is no significant difference across units (i.e. no panel effect).

**Table 7. Breusch and Pagan Lagrangian Multiplier(LM) test.**

| Panel A: Estimated Results for Random effects | | |
|---|---|---|
| | *Var* | *sd = sqrt(Var)* |
| TFR | 0.4345 | 0.6591 |
| e | 0.0104 | 0.1021 |
| u | 0 | 0 |
| Panel B: Results for Variance across Random and Pooled OLS model | | |
| *Test Summary* | *Chi Square Statistics* | *P value* |
| Var(u) = 0 | 0.00 | 1.0000 |

Estimated in Stata 13.

**Table 8. Panel estimates of Macro Data sample: (1990–2019).**

| Categories | Variables | FE | Pooled OLS | RE |
|---|---|---|---|---|
| | Constant | 9.439*** | 18.627*** | 18.627*** |
| | | (3.05) | (12.64) | (12.64) |
| *Empowerment* | Gross female secondary enrolment rate(GFSSER) | 0.005** | -.003*** | -.003*** |
| | | (2.35) | (-2.90) | (-2.90) |
| | LReal GDP | -0.142 | -0.031 | -0.031 |
| | | (-1.22) | (-0.46) | (-0.46) |
| | Female Labour force Participation rate(LFPRF) | -0.014*** | 0.009*** | 0.009*** |
| | | (-4.60) | (4.95) | (4.95) |
| *Health* | Infant Mortality rate(IMR) | 0.009 | 0.014*** | 0.014*** |
| | | (1.21) | (4.61) | (4.61) |
| | Contraceptive Prévalence rate(CPR) | -0.064*** | -0.035*** | -0.035*** |
| | | (-10.89) | (-11.73) | (-11.73) |
| *Connectivity* | Urban population(LUP) | -0.142* | -0.240*** | -0.240*** |
| | | (-1.52) | (-4.08) | (-4.08) |
| | Mobile subscriptions(MOBSUB) | 0.0007 | -0.0007 | -0.0007 |
| | | (1.28) | (-1.52) | (-1.52) |
| | Female Population(FEMPOP) | 0.071 | -0.203*** | -0.203*** |
| | | (1.22) | (-5.80) | (-5.80) |
| *Cost of Living* | Consumer price index(inflation)(LCPI) | -0.413*** | -0.118*** | -0.118*** |
| | | (-6.47) | (-3.01) | (-3.01) |
| *Additional statistics* | R-bar Squared | 0.93 | 0.96 | 0.87 |
| | F-statistic | 222.45*** | 446.61*** | 4019.1*** |
| | Observation | 180 | 180 | 180 |
| | rho | 0.98 | - | 0 |
| | corr(u_i, X) | -0.598 | - | 0 |

*Note*: ***, **,* indicates significance at 1%, 5% and 10% respectively.

Estimated in Stata 13.

Here, the p value is greater than 0.005 and thus, we fail to reject the null and conclude that random effects is not appropriate. There is no evidence of significant differences across countries, therefore we can run a simple Pooled OLS regression.

**3.2.2. Panel estimates.** As the Hausman test has eliminated random effects model as the most suitable model, and Breusch and Pagan Lagrangian Multiplier has also refused random effects model, we select with confidence now that fixed effects model will yield the best estimates, followed by Pooled OLS. Random effects model is retained in the estimation to compliment/compare results of fixed and OLS estimates. The panel estimates of Macro Data using 3 methods is given in Table 8.

The FE model (most favoured in linear panel methods) indicates some interesting results and almost all (except secondary enrolment) signs of the variables are as expected and in line with the theoretical expectations. The discussion for each of the categories of variables is given below.

**a. Empowerment**

Females' secondary school enrolment rate, as a measure of women's empowerment has a positive and significant impact on fertility under fixed effects, while being negatively associated with fertility under OLS/Random model, which is theoretically correct, and similar to studies

done by Ali & Gurmu [40]. The reason for positive correlation could be that progress in gender parity in education is somewhat disrupted in the Pacific. Furthermore, incremental changes to female gross enrolment rate are low and may not significantly affect fertility decisions. Even though the coefficient size for enrolment is not that substantial, women's education level has a bigger impact on the fertility rate than men.

Real GDP in the form of income is an important fertility determinant as well. Real GDP is negatively correlated with fertility under all three models; however, it is not statistically significant. The estimates under fixed effects yield the largest negative coefficient. A 1% increase in Real GDP will cause 0.14% decrease in fertility. The possible reason why Real GDP is classified as an insignificant factor could be due to the fact that PICs has not yet achieved conditional convergence and their Real GDP are still growing annually, thus the values for Real GDP for Pacific has not reached a threshold that could significantly influence fertility. The results obtained are similar to studies done by Becker [41].

Labour force participation rates (except the second and third estimation models), in line with the theory, have negative signs. In particular, the women labour force participation rate emerges as the most important determinant of fertility out of all other empowerment indicators, at 1% significance level. This result is similar to research done by Galor & Weil [42]. A 1% increase in women's labour force participation decreases fertility by around 0.014%. Pacific women who participate in the labour force have to sacrifice more in terms of foregone wages for childcare (assuming women are the basic childcare providers). Thus, women's labour force participation rate has an inverse relationship with fertility rate.

### b. Health

Infant mortality rate under all three models yields positive sign and except for fixed effects, infant mortality rate plays a significant role (1% significance level) in influencing fertility rates, which is theoretically correct. Generally, couples had more children in the past as a measure to ensure that at least one child survived to a later age. Infant mortality rate was high in Pacific in the 1900's because of underdeveloped heath care system and inadequate health professionals to look after the people. Due to this, treatment of some illness/disease was not possible/effective, which led to increase in child deaths. Hence, couples had more children to ensure survival hood of some, therefore the positive correlation [43]. Alternatively, as time progressed, massive structural change caused improvement in health and medical services, ensuring long life for newborn, thus over time, as infant deaths decreased, so did fertility rate. The fixed effects model correctly captures this significant shift in variables, causing IMR to become insignificant for fertility determination.

Contraceptive prevalence emerges from our all estimations as an important determinant of fertility. Intuitively, one expects a strong relation between contraceptive usage and fertility. All the estimations point increase in contraceptive usage lowers fertility by 0.05% on average. Even though contraceptive prevalence rate has increased generally, it is still lacking in most PICs, hence to effectively promote effective contraceptive use, policies and programs must address both the demand and supply side. Effort should include school-based and community-level education to improve knowledge of pregnancy risk, increase awareness of the importance of contraceptive use for the health of women and the well-being of the family, and dispel misconceptions about the adverse health consequences of contraception. Interventions should also promote communication between partners to ensure proper understanding of each other's position on fertility intentions and contraception [44].

### c. Connectivity

The sign for urban population under all three models is negative and statistically significant at 10% for fixed effects and at 1% for both Pooled OLS and random effects. In general,

urbanization is thought to lower fertility since living in a city would presumably raise the costs of raising children. In comparison to rural regions, urban housing is more expensive, and children are presumably less beneficial in household output [45]. In addition, urbanization may be linked to ideational transformation, or the changing of ideas and attitudes about big families. Furthermore, city dwellers in Pacific may have easier access to contemporary birth control, allowing them to more successfully implement any wish to decrease children. All the estimations point increase in urbanisation lowers fertility at least by 0.14%.

Theoretically, increase in mobile subscription enhances people's connectivity that could affect fertility in a positive way. Access to mobile phones, along with the internet, may divert young people's mind, especially teenagers, into engaging in sexting or viewing explicit contents. This may influence them to engage in sexual activities, which could lead to unwanted teenage pregnancy. Estimations under fixed effects proves this whereby the coefficient of mobile subscription is positive (but lower coefficient of 0.0007), and is not statistically significant. All other estimations yield negative and insignificant coefficients. Alternatively, increase in access to sexual and reproductive rights and higher education of youths in the Pacific overtime might suppress the negative effects of mobile phone usage, causing it to become an insignificant indicator of fertility.

Theoretically, higher female population in a country would mean higher gross reproductive rates, assuming all those females born are fertile, will get married in the future and will have a child. Estimations under fixed effects model show this whereby the coefficient of female population is positive, but not statistically significant. All other models yield negative and significant coefficients at 1% level. However, sterility is also a global concern. In the less developed countries, statistics show that only 50% of those affected with sterility (or related issues) seek interventions due to financial, medical, and cultural constraints [46]. For this reason, a rise in female population would not necessary mean a rise in fertility because of confounding factors such as sterility. Hence, a rise in female population as a factor influencing fertility becomes insignificant. Furthermore, given the fact that women's rights are now realised in the Pacific and they are given equal chance for education and employment, this may cause negative association with fertility. This is because educated and working female population would have delayed marriage and delayed child bearing, thus supressing fertility.

### d. Cost of Living

An increase in cost of living normally discourages people from engaging in those activities that might add up to the existing pile of costs. Individuals engage in cost-benefit analysis and normally pursue those activities that give them the highest utility or economic benefit. Consistent with the estimates of Becker [41], all our 3 models result show a strong negative and significant correlation between CPI (Inflation) and fertility rate, at 1% significance level. Under fixed effects, 1% increase in CPI (inflation) will decrease total fertility rate by 0.41%. This is a very significant relationship and can be seen as the strongest indicator of fertility in the Pacific. Rapid changes to the price of basic goods and services decrease people's welfare. Given that Pacific is a consumption driven society with limited to no savings, increase in basic cost of living will make people think twice of having more children, as their ability to raise/nurture them decreases with fluctuating expenses.

## 4. Conclusion and policy implications

This study was undertaken to determine factors influencing fertility in the 6 selected PICs. A balanced macro-level data, obtained from World Banks Development Indicators and World Health Organisation Database ranging from 1990–2019, was used for analysis. Using the most

preferred fixed effects for estimation and random effects/ Pooled OLS to complement and compare the main results, we find that female labour force participation rate, because of increase in education; contraceptive prevalence and CPI (inflation) are the key factors causing decrease in total fertility rate, at 1% significance level. Other variables such as Real GDP and urbanisation are also negatively correlated with total fertility rate, however the significance of GDP is not realised while urbanisation is significant at 10% under fixed effects and at 1% under Pooled OLS and random effects. Infant mortality rate, mobile subscription and female population are positively correlated with fertility rate, while its significance only being realised under random and pooled OLS model (except mobile subscription) at 1% level.

The findings implicate that while some variables might be affect fertility (either positive or negative), one has to deal with the fact that it is the current significance of the variable that shapes the future behaviour of that particular variable/regressor with fertility rate. The results show that variables such as labour force participation rate, contraceptive prevalence rate and CPI as a measure of inflation, are highly significant at 1% level. Looking at the individual trend of each variable, we note that contraception use in the Pacific is increasing over time and so is the female labour force participation rate, due to increase in female education. This trend implies that continuous increase in these variables would have a detrimental impact on fertility, and this change is non-discretionary, because of global diffusion process. Real GDP, which is negatively linked with fertility, is seen to be insignificant. This is because many PICs GDP is not at a substantial level to influence fertility. However, it is expected that with time, Real GDP will continue to expand indefinitely until it reaches its steady state (convergence). CPI as a measure of inflation on the other hand is something that can be managed or controlled by government through various fiscal policies or through central bank's monetary policies. Thus, given the continued fertility decline because of socio-economic and demographic changes over time, female employment, contraceptive use and Real GDP are variables that will drive individual countries fertility to replacement level in the future. This is because time trend is attached to these variables, which the government has no control over. All other variables such as urbanisation, female population, mobile use and infant mortality theoretically correlate with development in income and education variables, causing modernisation and improvement in health service. Additionally, fertility affects population growth more than mortality and migration, therefore causal relationship would exist between indicators of population growth and indicators of development [47], as illustrated in Fig 2.

Fig 2 shows the dynamics of population and development—the balance between availability and consumption of resources. As societies develop, the complexity and competition for resource increase. The result of development is the expansion of per capita resources and historically, the growth of per capita output has exceeded per capita growth in consumption. Thus, each additional person in the population contributes to both production and consumption. Therefore, population growth and development work both ways and either of them can reinforce each other.

In light of rapid fertility decline and a possible decrease in output in future due to shortage of labour force, government would find it difficult to initiate policies in the short run that would increase fertility because variables causing rapid fertility decline are driven independently and government policies would be ineffective in that stance. Hence, given the current trends and expected future developments of key variables such as female employment, contraceptive use and GDP, the authorities need to start streamlining their policies to ensure that fertility does not drop to a level in the future that could cause serious shortage of labour force and productivity.

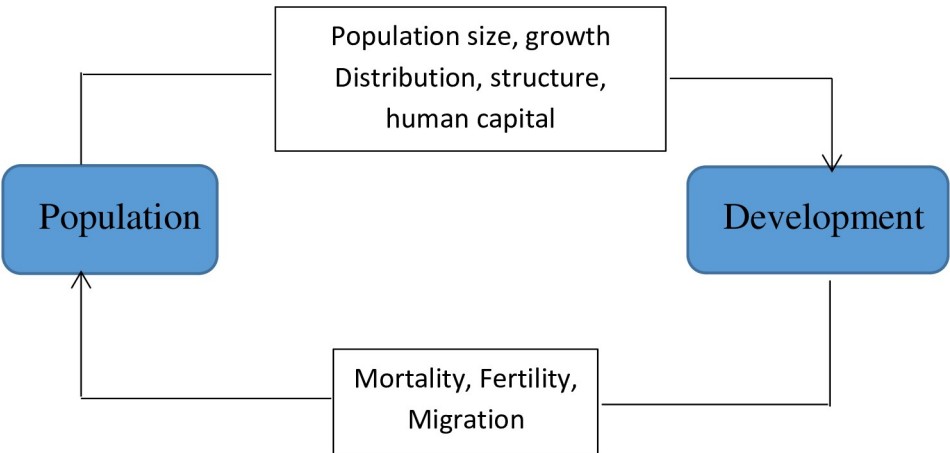

**Fig 2. Relationship between population growth and development.** Gould, W. (2009). Population and development. London; New York: Routledge [47].

## 5. Limitations

This study is based on several approximations and assumptions, thus recognising potential weak points of this research become important. Given that this research is drawn from works of Davis and Blake [24] and Bongaarts [25], it has employed several demographic and socio-economic variables that have been prescribed in their study. However, important variables such as frequency of sexual intercourse, induced abortion, mean age at marriage, family planning and Postpartum infecundability have not been included in this study. The reason is that Pacific countries dataset is still underdeveloped. Data on variables, such as those mentioned above are not collected at a regular interval because of the inconsistency in conducting demographic and health surveys at every 5-year interval, causing delay in data build-up process. The inclusion of the above listed variables would have benefited the study a lot, in terms of better understanding biological factors influencing fertility in the region.

## Supporting information

**S1 Data.**
(XLSX)

## Author Contributions

**Conceptualization:** Sumeet Lal, Rup Singh, Keshmeer Makun, Nilesh Chand.

**Data curation:** Sumeet Lal, Rup Singh, Nilesh Chand.

**Formal analysis:** Sumeet Lal, Rup Singh, Keshmeer Makun.

**Investigation:** Sumeet Lal, Rup Singh.

**Methodology:** Sumeet Lal, Rup Singh, Keshmeer Makun.

**Software:** Sumeet Lal, Rup Singh.

**Supervision:** Sumeet Lal, Rup Singh.

**Validation:** Sumeet Lal, Rup Singh.

**Visualization:** Sumeet Lal, Rup Singh, Mohsin Khan.

**Writing – original draft:** Sumeet Lal, Rup Singh.

**Writing – review & editing:** Sumeet Lal, Rup Singh, Keshmeer Makun, Nilesh Chand, Mohsin Khan.

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
