## [Decision Letter · Decision Letter 0]

11 Jun 2021

PONE-D-21-02736

Determinants of Fertility Rates in Selected Pacific Islands Countries: A Multi-Level Empirical Study

PLOS ONE

Dear Dr. Lal,

Thank you for submitting your manuscript to PLOS ONE. After careful consideration, we feel that it has merit but does not fully meet PLOS ONE’s publication criteria as it currently stands. Therefore, we invite you to submit a revised version of the manuscript that addresses the points raised during the review process.

The topic covered in the article could be an interesting read for researchers. The manuscript attempts to explore the nuances of various indicators on fertility in selected six Pacific Islands countries. The manuscript is excessively lengthy and severely lacks coherence in both the introduction and discussion section. For example, sentences between lines number 68-94 are poorly written and can be considered for deletion. Similarly, at various places in the manuscript, authors have failed to maintain consistency and coherence in the sentences. 

The acronyms used in the analysis need to be expanded when introduced in the manuscript—besides, the authors need to seriously check the in-out referencing. For example, WDI (2017) and Stock and Watson (2008) are not listed in the reference list. This article falls short on various grounds. 

In the process of empirical assessment, a range of statistical models have been applied in this manuscript; however, it falls short in terms of argument building and justification on the choices of these econometric models and, most importantly, in referencing. The review of the analysis in the study are as follows,

The methodology section of the manuscript is very confusing and needs serious attention. The authors have mentioned that both primary and secondary data sources have been carried out in the study. However, details on primary data source such as selection of households, sample adequacy, sampling design, statistical power are entirely missing from the analysis. The authors have selected 30 random samples from each country; however, in the study, N is equal to 700. The sample from each county is mentioned as 140. This creates massive confusion on the validity of the analysis.Who were the respondents in the survey? All women of reproductive age group 15-49 years or currently married women of the reproductive age groups. It would be suggested to consider rewriting the methods section in a more detailed manner.The authors have explained a range of fertility theories in the literature review. However, in the analysis section, they have adopted indicators based on Davis and Blake (1956) and Ahlburg and Cassen (1993) to conduct analysis. Why so ? However, the later study finds no place in the literature review.Another observation from the analysis is that the availability of knowledge of contraceptives by women is considered a proxy of the CPR for the micro-level data. Surprisingly, the authors did not ask whether the women were currently using contraceptives or not? Why so ?The authors need to bring more clarity on the indicators chosen for the study. For example, what a reader should comprehend from the “Highest level of educational attainment of women”.  Instead, in the table “Level* of education”* variable is mentioned. Are these two variables same ? Similarly, what is the meaning of “unmet need of pregnancy” ? does refer to failure in conceiving due to primary or secondary sterility or infertility, infecundity, or simply a mismatch between actual and desired fertility levels. Consider using the name for the indicators, which do not create confusion to the reader.It is highly recommended to provide frequency distribution or descriptive statistics against each explanatory variable so that readers can comprehend sample adequacy for any empirical analysis.There are some shortcomings in describing the stepwise structure of the econometric models. What made the authors run these three OLS models on the survey data is absent in the study. The motivation of using different models should be underlined in the method section as to what hypothesis authors are attempting to testify through each model. The variable “the number of kids to the women” depends on the age of the women. The fertility levels are dependent on the age at marriage and exposure to future fertility. How authors have accounted for the offset while dealing with OLS model. Also, considering the nature of the dependent variable, why do authors not employ the Poisson regression model on microanalysis?The authors have employed the GMM estimation procedure assuming endogeneity in the model. To substantiate the premise, authors have cited study carried out by Arellano and Bond (1991). However, if such endogeneity persists or not in the case of PIC countries has not been empirically testified. I would suggest authors perform either the Hausman test or the Durbin-Wu-Hausman test to assess endogeneity in the present study's context.Since the study is based on several approximations and assumptions; therefore, authors must include a section on ‘Limitation of the study’.Lastly, the authors have failed to justify the study's title in the abstract, rationale and discussion section.Considering grammatical errors and spelling mistakes in the manuscript, an immediate suggestion would be to get this manuscript copy-edited by some professional editor. 

We look forward to receiving your revised manuscript.

Kind regards,

Srinivas Goli, Ph.D.

Academic Editor

PLOS ONE

Journal Requirements:

2. Please improve statistical reporting and refer to p-values as "p<.001" instead of "p=.000". Our statistical reporting guidelines are available at https://journals.plos.org/plosone/s/submission-guidelines#loc-statistical-reporting

Additional Editor Comments:

The topic covered in the article could be an interesting read for researchers. The manuscript attempts to explore the nuances of various indicators on fertility in selected six Pacific Islands countries. The manuscript is excessively lengthy and severely lacks coherence in both the introduction and discussion section. For example, sentences between lines number 68-94 are poorly written and can be considered for deletion. Similarly, at various places in the manuscript, authors have failed to maintain consistency and coherence in the sentences.

The acronyms used in the analysis need to be expanded when introduced in the manuscript—besides, the authors need to seriously check the in-out referencing. For example, WDI (2017) and Stock and Watson (2008) are not listed in the reference list. This article falls short on various grounds.

In the process of empirical assessment, a range of statistical models have been applied in this manuscript; however, it falls short in terms of argument building and justification on the choices of these econometric models and, most importantly, in referencing. The review of the analysis in the study are as follows,

1. The methodology section of the manuscript is very confusing and needs serious attention. The authors have mentioned that both primary and secondary data sources have been carried out in the study. However, details on primary data source such as selection of households, sample adequacy, sampling design, statistical power are entirely missing from the analysis. The authors have selected 30 random samples from each country; however, in the study, N is equal to 700. The sample from each county is mentioned as 140. This creates massive confusion on the validity of the analysis.

2. Who were the respondents in the survey? All women of reproductive age group 15-49 years or currently married women of the reproductive age groups. It would be suggested to consider rewriting the methods section in a more detailed manner.

3. The authors have explained a range of fertility theories in the literature review. However, in the analysis section, they have adopted indicators based on Davis and Blake (1956) and Ahlburg and Cassen (1993) to conduct analysis. Why so ? However, the later study finds no place in the literature review.

4. Another observation from the analysis is that the availability of knowledge of contraceptives by women is considered a proxy of the CPR for the micro-level data. Surprisingly, the authors did not ask whether the women were currently using contraceptives or not? Why so ?

5. The authors need to bring more clarity on the indicators chosen for the study. For example, what a reader should comprehend from the “Highest level of educational attainment of women”. Instead, in the table “Level of education” variable is mentioned. Are these two variables same ? Similarly, what is the meaning of “unmet need of pregnancy” ? does refer to failure in conceiving due to primary or secondary sterility or infertility, infecundity, or simply a mismatch between actual and desired fertility levels. Consider using the name for the indicators, which do not create confusion to the reader.

6. It is highly recommended to provide frequency distribution or descriptive statistics against each explanatory variable so that readers can comprehend sample adequacy for any empirical analysis.

7. There are some shortcomings in describing the stepwise structure of the econometric models. What made the authors run these three OLS models on the survey data is absent in the study. The motivation of using different models should be underlined in the method section as to what hypothesis authors are attempting to testify through each model.

8. The variable “the number of kids to the women” depends on the age of the women. The fertility levels are dependent on the age at marriage and exposure to future fertility. How authors have accounted for the offset while dealing with OLS model. Also, considering the nature of the dependent variable, why do authors not employ the Poisson regression model on microanalysis?

9. The authors have employed the GMM estimation procedure assuming endogeneity in the model. To substantiate the premise, authors have cited study carried out by Arellano and Bond (1991). However, if such endogeneity persists or not in the case of PIC countries has not been empirically testified. I would suggest authors perform either the Hausman test or the Durbin-Wu-Hausman test to assess endogeneity in the present study's context.

10. Since the study is based on several approximations and assumptions; therefore, authors must include a section on ‘Limitation of the study’.

11. Lastly, the authors have failed to justify the study's title in the abstract, rationale and discussion section.

12. Considering grammatical errors and spelling mistakes in the manuscript, an immediate suggestion would be to get this manuscript copy-edited by some professional editor.

Reviewers' comments:

Reviewer's Responses to Questions

**Comments to the Author**

1. Is the manuscript technically sound, and do the data support the conclusions?

Reviewer #1: Partly

2. Has the statistical analysis been performed appropriately and rigorously? 

Reviewer #1: Yes

3. Have the authors made all data underlying the findings in their manuscript fully available?

Reviewer #1: No

4. Is the manuscript presented in an intelligible fashion and written in standard English?

Reviewer #1: No

5. Review Comments to the Author

Reviewer #1: The introduction is quite long and readers may struggle to understand the research problem the authors are trying to address. It is not clear the research question(s) and the new knowledge the study will contribute

6. PLOS authors have the option to publish the peer review history of their article (what does this mean?). If published, this will include your full peer review and any attached files.

Reviewer #1: No

---

## [Author Response · Author response to Decision Letter 0]

31 Aug 2021

Reviewer 1 response

Author response 

We really appreciate the reviewers for their valuable comments. We have now embedded the necessary changes to our paper and the changes have been identified below. 

Introduction is now conscise and it clearly addresses the research questions and new knowledge that this study contributes. The hypothesis being tested is explained in introduction, with a more details being included under methods sections, along with the reasons variable & model selection. More details on the data sources, including data quality and limitations has been included in methods section, as well as a tabulated version of socio-economic and demographic context of the six countries. The results and discussion section are now easily distinguishable. 

Data sources, sample size, and the main analysis model used in the analysis has been indicated

The sign for the main variables, as well as their significance level has been stated.

This comment has been now addressed as part of the conclusions. 

The entire introduction has been re-written and reproduced statistics have been either dropped or presented as a summary in table. 

The objective of the study has now been made clear in the text. 

Conceptual framework has been brefly outlined towards the end of introduction, along with the key hypothesis. More detailed version of hypothesis can be found under methods

The 6 country contexts have been summarised into a table

Ln 141 on outdated research has been now identified and stated in text.

This statement has been completely removed

It has been removed

Micro-analysis has been removed, as per the reviewers suggestion and the justification of macro-level data is provided in introduction and also under methods section.

Survey of literature has now been summarized and merged with introduction

This has been addressed in the form of table

This has been now accounted for under data sources and limitations section

Now only Macro-level dataset is used, along with its justification

This has been now explained under model specification section. Frameworks of Davis & Black(1956) and Bongaarst(1987) guided the variable selection

This has been indicated under data sources and limitations section

Now only Macro-level dataset is used, as per the suggestions from the reviewer.

Household level analysis has been removed from this study, details now not needed. 

Household level analysis has been removed from this study – we do not need these anymore.

All relevant research ethics were followed for obtaining and using household data. Now since micro analysis are removed,the approval (although we have obtained) is not required. 

The net effect of the predictors on the outcome variable is examined by removing the influence of those time-invariant features using FE.

This has been now included under estimation techniques

Micro-analysis has been removed and this section has been redone, with appropriate re-estimations and additional statistics being included in the analysis. The results and discussion section are now easily distinguishable. The implication has been stated in brief under findings, but has bene comprehensively covered under conclusion and implications section

This has been now explained

This sentence has been removed

This sentence has been removed

Reviewer 2 response

The manuscript has been reduced in size and introduction and discussion section has been made concise.

Sentences between lines number 68-94 has been deleted.

All those acronyms used in the analysis has been expanded and reference list has also been updated

The micro-level analysis has been completely removed from the study. This study now only has macro-level analysis of 6 PICs, for which the data ranges from 1990-2019. This makes this section much more clear as suggested by reviewer 1.

Primary data is now not used, Initially, this section benefitted from repeated sampling method available in STATA. Now primary data analysis is removed.

The micro-level analysis(survey) has been completely removed from the study

Fertility theories has been removed from literature review. 

Indicators suggested by Davis and Blake (1956) has been used in this study as it offered a quantitative framework to decompose fertility into its proximate determinants (social and economic variables). This framework is useful for performing comparative fertility analysis. Ahlburg and Cassen (1993) later draws work from Davis and Blake(1956). Both of these study has now been properly cited in the text 

The question was asked. However, the micro-level analysis(survey) has been completely removed from the study

The indicators used for macro-level studies has now been clearly identified and explained. All the variables associated with micro analysis, as well as the micro level estimation has been removed from this study, which now removes ambiguity. 

Descriptive statistics against each explanatory variable, as well as the correlation matrix for macro variables has been now made available in the research

OLS models on the survey data has been removed. All the models used for macro analysis has been well described and justified, along with the key hypothesis being tested in the model

The micro-level analysis(survey) has been completely removed from the study

GMM estimation has been removed. Fixed, random and Pooled OLS has been employed in this study. Appropriate testing such as Hausman test and Breusch Pagan Lagrangian multiplier has been employed to access the endogeniety in the study and for selection of the appropriate models

Separate section on limitation of the study has been included in this paper

The title has been changed. Since this study has employed socio-economic and demographic variables, the title has been renamed to “Socio-Economic and Demographic determinants of Fertility in Six Selected Pacific Islands Countries: An Empirical Study”. The title has been also justified in abstract and discussion section

The entire article has been proof read again to check for any anomalies.

---

## [Editor Report · Decision Letter 1]

7 Sep 2021

Socio-Economic and Demographic Determinants of Fertility in Six Selected Pacific Islands Countries: An Empirical Study

PONE-D-21-02736R1

Dear Dr. Lal,

We’re pleased to inform you that your manuscript has been judged scientifically suitable for publication and will be formally accepted for publication once it meets all outstanding technical requirements.

Kind regards,

Srinivas Goli, Ph.D.

Academic Editor

PLOS ONE

Additional Editor Comments (optional):

Authors revised the paper according to reviewers comments, thus I am recommending this paper.
---

## [Editor Report · Acceptance letter]

13 Sep 2021

PONE-D-21-02736R1 

Socio-economic and demographic determinants of fertility in six selected Pacific Island countries: An empirical study 

Dear Dr. Lal:

I'm pleased to inform you that your manuscript has been deemed suitable for publication in PLOS ONE. Congratulations! Your manuscript is now with our production department. 

Kind regards, 

on behalf of

Dr. Srinivas Goli 

Academic Editor

PLOS ONE